# Dataset on Force Myography for Human–Robot Interactions

Umme Zakia [1] and Carlo Menon [1,2,*]

1    Menrva Research Group, School of Mechatronic Systems Engineering and Engineering Science, Simon Fraser University, Metro Vancouver, BC V5A 1S6, Canada

2    Biomedical and Mobile Health Technology Laboratory, Department of Health Sciences and Technology, ETH Zurich, Lengghalde 5, 8008 Zurich, Switzerland

\*    Correspondence: cmenon@sfu.ca or carlo.menon@hest.ethz.ch; Tel.: +1-778-782-9338; Fax: +1-778-782-7514

**Abstract:** AbstractForce myography (FMG) is a contemporary, non-invasive, wearable technology that can read the underlying muscle volumetric changes during muscle contractions and expansions. The FMG technique can be used in recognizing human applied hand forces during physical human robot interactions (pHRI) via data-driven models. Several FMG-based pHRI studies were conducted in 1D, 2D and 3D during dynamic interactions between a human participant and a robot to realize human applied forces in intended directions during certain tasks. Raw FMG signals were collected via 16-channel (forearm) and 32-channel (forearm and upper arm) FMG bands while interacting with a biaxial stage (linear robot) and a serial manipulator (Kuka robot). In this paper, we present the datasets and their structures, the pHRI environments, and the collaborative tasks performed during the studies. We believe these datasets can be useful in future studies on FMG biosignal-based pHRI control design.

**Keywords:** force myography technique; interactive applied hand forces in dynamic motion; physical human-robot interactions; FMG-based pHRI

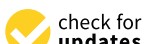



## 1. Summary

In industrial physical human–robot interactions (pHRI), a human worker mainly interacts using hand forces to perform a collaborative task. Commercially available force sensors attached to the robot can read the applied force, which requires the worker to apply force on the sensor and confines the worker's movement within the workspace. On the other hand, measuring human-applied force via wearable sensors can be advantageous in allowing unrestricted movements while including human bio feedback in the control loop. Among many wearables, force myography (FMG) is a contemporary, non-invasive, affordable technique, that can map exerted forces from muscle contractions via force sensing resistors (FSRs) [1,2]. An FMG band wrapped around upper extremities can read the voluntary muscle changes during isometric hand force, grasping force or interactive applied hand force during dynamic arm movements [3–10].

In the literature, studies conducted on human–machine interactions (HMI), or human–robot interactions (HRI) are mainly monitored using vision system, proximity detectors, wideband/radio frequencies, and invasive or non-invasive sensory systems. Only few HRI datasets are publicly available, such as: PinSoRo [11] and DREAM [12] for understanding social constructions of childhood; CADDY [13] and Aquaticus [14] for under water interactions with machines; MAHNOB-HCI [15] for understanding emotions; MHHRI [16], P2PSTORY [17] and UE-HRI [18] for understanding behavioral and socio-emotional profile; and TacAct [19] for tactile information during interactions with robots.

These datasets contain video and audio recordings, image depth sensing, skin temperature, eye-gaze tracking, physiological sensors, tactile sensors and acceleration sensors.

Including human biosignals to control or interact with machines is an ongoing research area. Many studies used biosignals such as electroencephalogram (EEG), electrooculogram (EOG) and electromyogram (EMG) for neuronal association with machines. These biosignals are mostly acquired from a specialized tissue, organ, or the nervous system. Research has shown that including human feedback via one or more of these biosignals was feasible [20–26]. However, implementing human–machine interactions via these signals is sometimes impractical because of poor signal quality, bulky signal processing equipment, and restrictive human movements. Alternatively, many HRI studies [27–34] utilized the traditional, non-invasive, wearable surface electromyography (sEMG) technique, which measures the electrical activities of underlying muscle bundles. However, none of the myography-based HRI datasets are publicly available. Interestingly, the FMG technique was found comparable with the conventional sEMG technique in hand gesture recognition, rehabilitation, or prosthetic control applications for human–machine interaction studies. Research has shown that the FMG technique was advantageous compared to sEMG technique, with lower cost, smaller signal processing units using Bluetooth technology, ease of wearing the band and, hence, a better choice for HMI [35–37]. Recently, FMG-based human–robot collaboration (HRC) tasks were conducted where an industrial YUMI robot avoided collision with a human participant by recognizing intentional or random hand movements [38]. In separate HRC studies, grasping forces via FMG data were used to recognize the intended tool (object) grasped by the human worker during a shared task [39,40].

Very recently, a few FMG-based HRI studies were conducted by the authors of this article, where human-applied interaction force in dynamic motion was predicted to perform a collaborative task. There is no other similar research work in predicting human interactive forces for HRC tasks like these studies because of the uncertain and dynamic HRI environment and availability of research funding and trained personnel. Understanding human intended intentions of interactions can be vital in developing safe collaborations between a human worker and a manipulator. Hence, this paper releases a dataset under the CC BY NC ND 4.0 license collected during the studies conducted in [6–10] while dynamic interactions happened between several human participants with a biaxial stage and a Kuka robot. These experimental data were collected using certain setups. A participant interacted with the robot, applying hand forces in several dynamic arm movements that represented common activities during human–robot interactions. The FMG bands wrapped around the forearm, or the upper arm, captured the muscle readings of a participant during an activity with the robot and were mapped to an applied force in a certain motion via a trained model. So far, to our knowledge, there is not a publicly available FMG dataset on pHRI application. Hence, this dataset will provide a starting point for future research works and avoid the need for collecting data. These data will also help researchers to understand the nature of the sensory signals that captured muscle activities during certain pHRI interactions. Our goal is to provide insight about the FMG signals and their applicability as a safety band in human–robot interactions, and inspire other researchers to work on this dataset. As the studies on FMG-based pHRI have revealed its viability [6–10], we hope that this release will aid to fill the gap in available datasets and to facilitate future research in biosignal-based HRI control design.

## 2. Dataset Collection Instrumentations

This dataset contains FMG data collected from two separate pHRI setups where a participant interacted with (i) a biaxial stage (2-DoF linear robot), and (ii) a serial manipulator (7-DoF Kuka robot). FMG data were collected from: (a) upper arm and forearm positions during interactions with the biaxial stage, and (b) forearm position only when interacted with the Kuka robot. Each row of FMG data had a corresponding true force reading (N) from a 6-axis force-torque (FT) sensor.

Participants in these studies were healthy, right-handed and their average age was $33 \pm 8$ years. They acknowledged the study protocol and signed the consent forms as approved by the Office of Research Ethics at Simon Fraser University, Canada. In this repository, total 18 participants' data are presented. Each participant is masked with a subject id (SubID: S1, S2, . . . , S18) only, there are no personal data associated in this public release to identify them.

- **FMG Bands**

Data presented in this paper were collected using two custom-made wearable FMG bands worn on the upper arm and/or forearm muscle bellies during pHRI interactions, as shown in Figure 1. The bands were made of force sensing resistors (FSRs) whose resistances changed when muscles contracted. During interactions, resistances of these FSRs decreased when pressure increased (muscle contracted) and vice versa. Each of these bands had 16 FSRs (TPE 502C, Tangio Printed Electronics, Vancouver, BC, Canada), with a length of roughly 30 cm. Data acquisition devices from National Instruments (NI USB 6259 and 6341, National Instruments, Austin, TX, USA) were used to collect data from these bands at 10–50 Hz. The FMG data presented in the .csv files are the measured voltage drops across the voltage divider (10–20 k$\Omega$ base resistor) against each FSR. Better understanding on the FMG band can be found in [1]. Each row in a file corresponds to 10–100 ms of the time-series data based on the setup used.

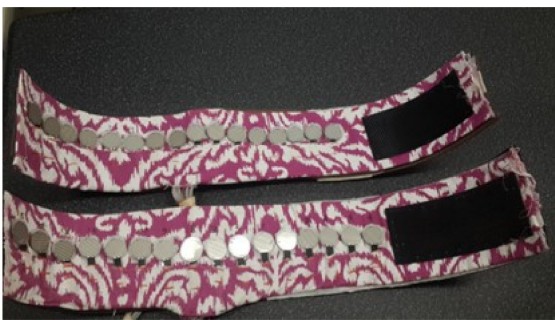 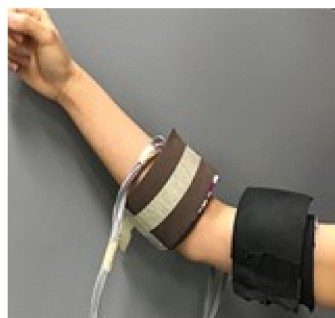

**Figure 1.** Two customized force myography (FMG) bands worn on the forearm and upper arm muscle belly of a participant to read muscle contraction.

- **The Biaxial Stage (2-DoF Linear Robot)**

This 2-DoF linear robot consisted of two linear stages (X-LSQ450B, Zaber Technologies, Vancouver, BC, Canada) for the desired translational movements of $450 \times 450$ mm travel distance in the X and Y plane. The bottom linear stage was placed in the X direction while the upper stage was placed in the Y direction, as shown in Figure 2. A customized 3D printed knob-like gripper was mounted on top of the biaxial stage. Implementing admittance control allowed compliant collaboration that enabled participants to grab the gripper and apply forces to slide it in any intended direction in real-time. A 6-axis FT sensor (NI DAQ 6210, National Instruments, Austin, TX, USA) was placed inside the gripper as the true label generator. Detailed information of this setup can be found in [6].

- **The Serial Manipulator (7-DoF Kuka Robot)**

The advanced KUKA LBR IIWA 14 R820 collaborative robot featured a 14 kg payload with an 820 mm reach. It came with built-in torque sensors in all joints except the end-effector and had its own controller: the 'Kuka Sunrise Cabinet'. Implementing torque control helped ensure compliant collaboration such that the displacements and trajectories of the robot were governed by applied hand forces realized on the end-effector of the robot.

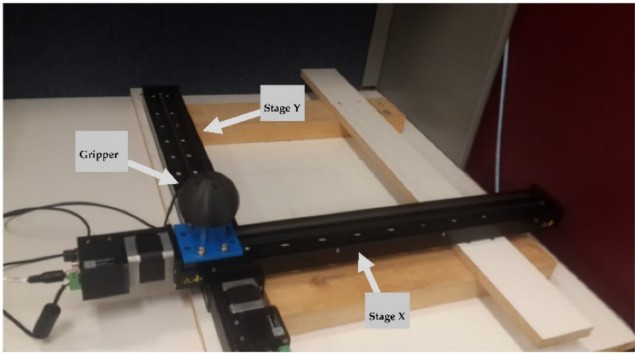

**Figure 2.** A biaxial stage with a knob-like gripper mounted on its top.

For pHRI, a custom-made cylindrical gripper was attached as the end-effector via a customized adapter. A 6-axis FT sensor (NI DAQ 6210, National Instruments, Austin, TX, USA) was placed between the gripper and the adapter for true label generation. The orientation of the gripper was kept at {0, pi, 0} for ease of grasping, as shown in Figure 3a.

For an object transportation task during human–robot collaboration (HRC), a 45 cm rectangular wooden rod was attached to the end-effector of the robot via a custom-made adapter, as shown in Figure 3b. The rod was parallel in the horizontal X dimension, with one end free to grasp and apply force to move it from point A to point B in the 3D plane. The 6-axis FT sensor was used as the true force label generator placed in between the adapter and the end-effector. Further details of both setups can be found in [9].

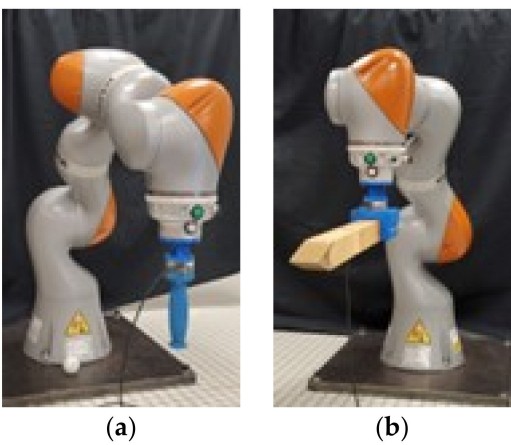

|  |  |
|:---:|:---:|
| (**a**) | (**b**) |

**Figure 3.** Kuka robot with different end-effectors for a participant to grab and interact with (**a**) a cylindrical gripper, and (**b**) a wooden rod.

## 3. Dataset Association

### 3.1. Dataset 1: pHRI between Human Participants and the Biaxial Stage

A total of 17 participants' (subjects') pHRI data with a biaxial stage are presented in Dataset 1 as 'pHRI_Biaxial_Stage'. Five different dynamic arm motions such as: "x-direction (X)", "y-direction (Y)", "diagonal (DG)", "square (SQ)", and "diamond (DM)" in the cartesian space were considered as the intended path trajectories for a participant to interact. During interactions, the participant, wearing upper arm and forearm FMG bands, grasped the gripper and continuously interacted for a certain time in sinusoidal fashion, as shown in Figure 4 and Table 1. Detailed study protocol and analysis are available in [6].

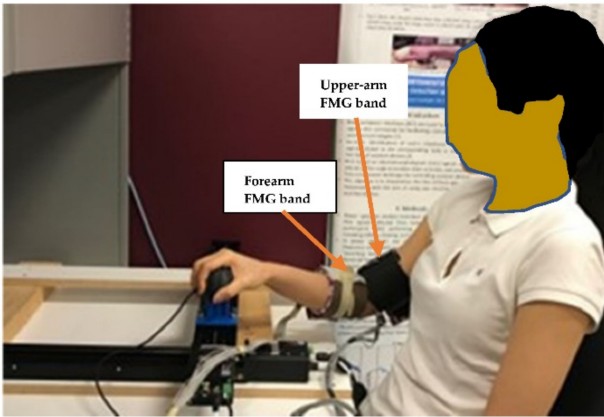

**Figure 4.** Participant wearing FMG bands on upper-arm and forearm interacts with the biaxial stage by grasping the gripper/knob.

**Table 1.** Five interactive arm motion patterns.

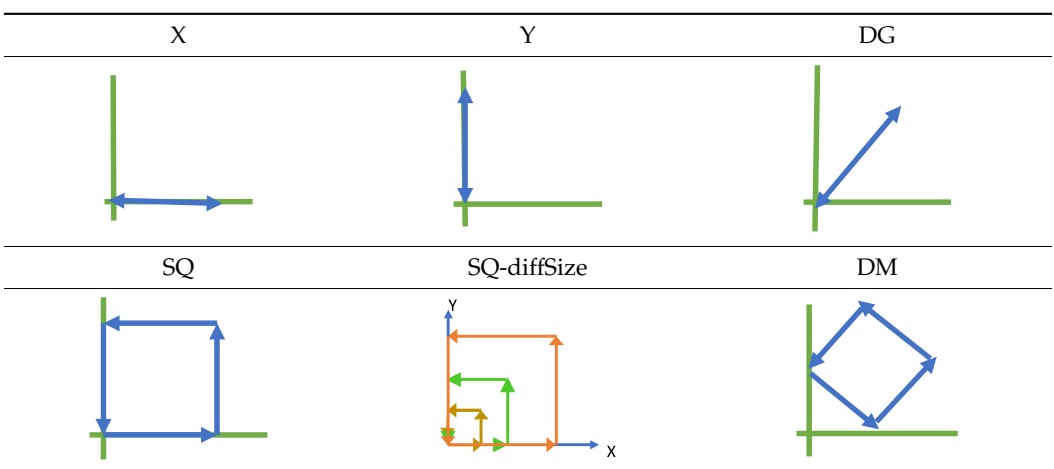

For model generalization, a multiple source dataset was constructed from participants S5, S6, S7, S8, and S10. The trained model was evaluated on repetitive participants: S3, S4, S9, and new participants: S11–S17. Detailed descriptions can be found in [7].

For domain adaptation and generalization, data were collected from participant S6 while interacting in square motion of different sizes: 'SQ-diffSize' [8]. For data collection and evaluation, interactions were performed continuously for a certain time in a sinusoidal motion on the planar surface, with directions as indicated with arrows in Table 1.

### 3.2. Dataset 2: pHRI between a Human Participant and a Manipulator

In this dataset, Raw FMG signals from a 16-channel forearm band are presented as 'pHRI_Manipulator'. It contains data during interactions between a participant S18 and a manipulator (Kuka robot) in 1D, 2D, and 3D planes. During pHRI, the Kuka robot had cylindrical gripper as the end-effector, as shown in Figure 5a [9,10]. The participant grabbed the cylindrical gripper, applied hand forces, and moved the gripper in 1D (X, Y, Z directions), 2D (XY, YZ, XZ planes) and 3D (XYZ plane). The robot followed the directions and trajectories of the human participant during interactions.

For collaborative tasks, a wooden rod was attached to the robot's flange. Participant S18 grabbed the open-end of the rod and moved it in the half-circle trajectories in the 3D plane from point A to point B, as shown in Figure 5b [9]. Detailed descriptions of these study protocols and data analysis are available in [9,10].

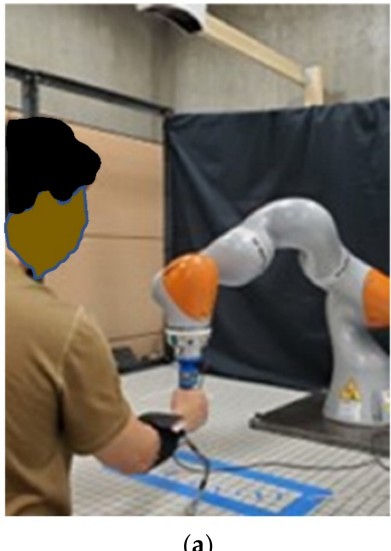 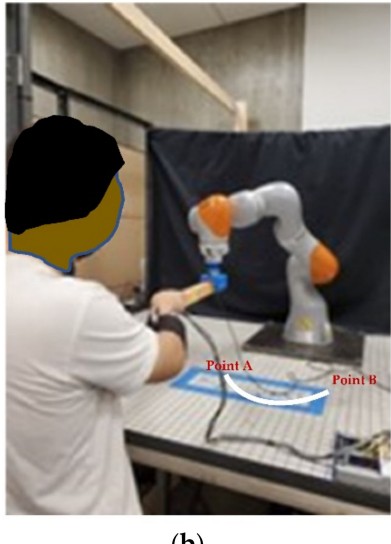

(**a**)                 (**b**)

**Figure 5.** (**a**) pHRI between participant S18 and the Kuka robot, and (**b**) HRC between participant S18 and the manipulator during moving a wooden rod in 3D workspace from point A to point B in a half-circular path.

## 4. Dataset Description

### 4.1. Dataset 1: pHRI_Biaxial Stage

Raw FMG signals from the 32-channel bands (two bands) along with true force label(s) are saved as .csv files with corresponding subject identification (SubID such as S1, S2, . . . , S17) and presented as tabular formats. For the first ten participants (S1–S10), 5 repetitions (rep0-rep4) of training data are included in this dataset for each arm motion (1D-X, 1D-Y, 2D-DG, 2D-SQ, and 2D-DM) collected during the study carried out in [6]. For 1D-X and 1D-Y directions, there are 33 columns in each data file, out of which the first column corresponds to the true force label ($F_x$ or $F_y$) in newtons (N). The positive force values (+N) indicate interactive forces in one direction while negative force values (−N) indicate the opposite direction because of continuous movements in sinusoidal fashion. The rest of the 32 columns are the 32-channel FMG data (32 feature space) collected from the upper arm and forearm bands. Data files on 2D-DG, 2D-SQ, and 2D-DM interactions have 34 columns, the first two columns of which are force readings from the FT sensor in the X and Y direction ($F_x$, $F_y$). The name format follows notation such as 'pHRI_BiaxialStage_S5_2D_DG_Rep3.csv' to indicate the 4th repetition of FMG data during interactions between the participant S5 and the biaxial stage in diagonal direction. A total of 100,000 × 34 data samples were collected during this study.

In the study conducted in [7], two repetitions (rep0-rep1) of interactive data in 1D-X and 2D-DG were collected from seven new participants (S11-S17) and were used to calibrate a long-term generalized model. As existing users, 2 repetitions of 1D-X and 2D-DG data were also collected from participants S3, S4, and S9. The naming convention for the repetitive participants are given as 'pHRI_BiaxialStage_S9_1D_X_Session2_Rep0.csv', with the additional tagging 'Session2' to indicate a second data-collection session. Each file has 400 rows of data. In addition to the existing dataset from the previous study [6], a total of 16,000 × 34 samples of data were collected for zero-shot learning.

The study conducted in [8] collected pHRI data between participant S6 and the biaxial stage in '2D-SQ-DiffSize' dynamic motion. A total of 16 repetitions of data collected in four separate sessions are included in the dataset. Data from the first three sessions (14 repetitions) were used for pretraining a deep learning model and the final session data were used for fine tuning [8]. These files have 600 rows and 34 columns, where the first two columns are the true force labels ($F_x$, $F_y$) and the rest are the FMG feature space distributions. A total of 9600 × 34 samples of data were collected for pretraining and finetuning. The

files have a naming convention like 'pHRI_BiaxialStage_S6_2D_SQ_diffSize_Rep0.csv' corresponding to the first repetition of interactions in 2D-SQ-diffSize motions with the biaxial stage. For model generalization, the pretrained model was evaluated on five participants (S1:S5) during interactions in '2D-SQ' motion [8]. Figure 6 shows the FMG signals capturing interactions with the biaxial stage in diagonal motion for a male and a female participant. The plot of these captured signals shows that the participants were interacting with the stage with their own pace of applied force.

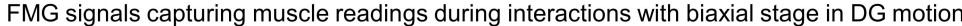

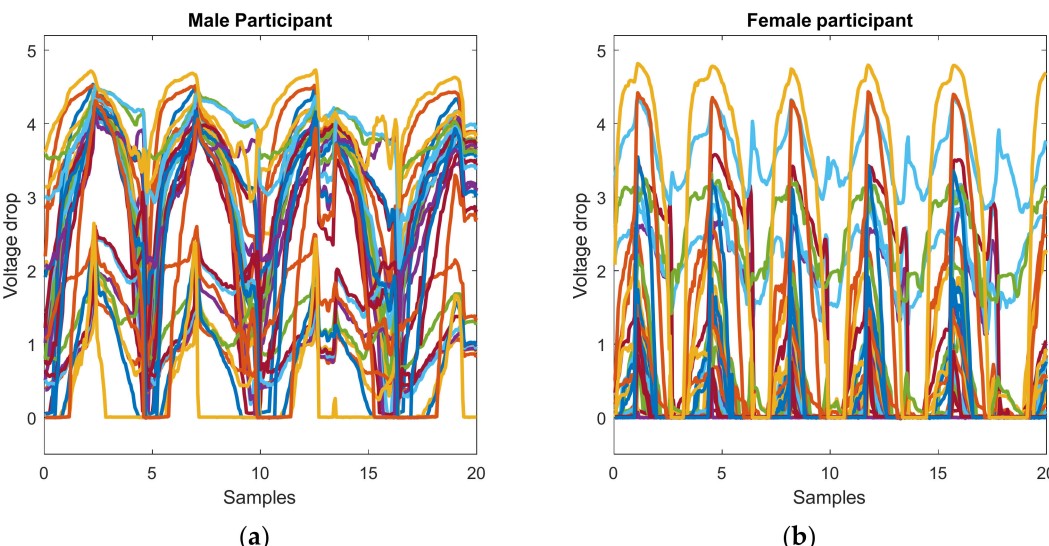

**Figure 6.** 32-channel FMG signals capturing muscle readings during interactions with biaxial stage in DG motion: (**a**) A male participant, and (**b**) A female participant.

### 4.2. Dataset 2: pHRI_Manipulator

All FMG data during interactions with the serial manipulator in the study conducted and presented in [9,10] are gathered in this dataset, saved as .csv files and presented in tabular formats. As before, the FT sensor data (true labels) have positive force values (+N), indicating applied force in one direction and negative force values (−N) in the opposite direction during interactions. For appropriate interaction data, 50 rows at the beginning and at the end can be stripped.

Five (5) repetitions (rep0-rep4) of data were collected for each dimension (1D, 2D, and 3D plane), and hence the naming format follows the notation starting with 'pHRI_Manipulator'. Each file has 19 columns, the first three of which are the true force labels ($F_x$, $F_y$, $F_z$) in newtons (N) and the last 16 columns of which are the 16-channel FMG features space. Data files have names such as 'pHRI_Manipulator_S18_1D_X_Rep0.csv' to indicate the first repetition of interaction data in 1D-X direction between the participant S18 and the manipulator. Likewise, 'pHRI_Manipulator_S18_2D_XZ_Rep4.csv', or 'pHRI_Manipulator_S18_3D_XYZ_Rep2.csv' file names mean final repetition of interactions between S18 and the Kuka robot in 2D-XZ plane or the third repetition in the 3D-XYZ plane. Figure 7 shows a few plots of the FMG signals during interactions with the Kuka robot in a certain direction such as 1D-X, 1D-Y, and 3D-XYZ.

For the collaborative task of moving the wooden rod in 3D, five repetitions of FMG data were collected and have names such as 'pHRI_Manipulator_S18_3D_XYZ_HRC_Rep3.csv', where 'HRC' means human–robot collaboration, 'Manipulator' is the Kuka robot, and the 4th repetition of interactive forces as participant 'SubID: S18' moved the rod collaboratively with the robot in the 3D plane.

Data repetitions collected during each category of pHRI are shown in Table 2.

Table 3 shows the summary of the dataset, format of each file, and the total repository presented.

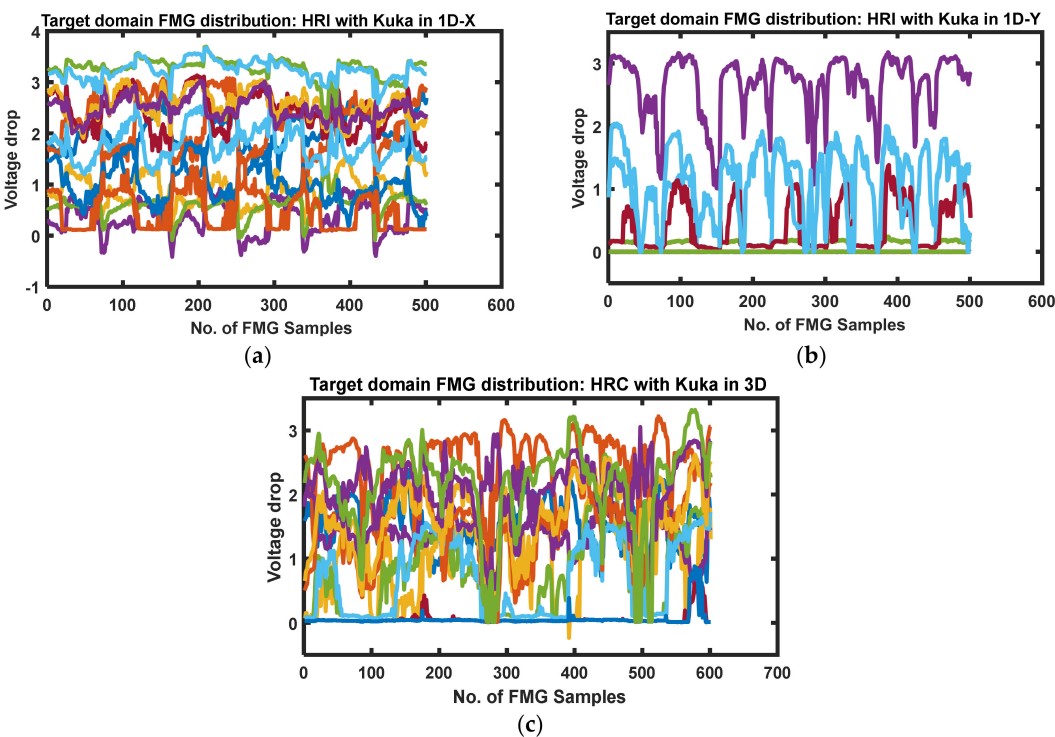

**Figure 7. 16 channel FMG signal readings during** pHRI between participant S18 and the Kuka robot: (**a**) in 1D-X, and (**b**) in 1D-Y, and (**c**) in 3D for HRC task. Adapted with permission from Ref. [9]. Copyright 2022 IEEE.

**Table 2.** Naming conventions followed.

| | Dataset 1: pHRI_BiaxialStage | | Dataset 2: pHRI_Manipulator |
|---|---|---|---|
| 1D-X | pHRI_BiaxialStage_SubID_1D_X_Rep0: Rep4.csv pHRI_BiaxialStage_SubID_1D_X_Session2_Rep0: Rep1.csv | 1D-X | pHRI_Manipulator_1D_SubID_X_Rep0: Rep4.csv |
| 1D-Y | pHRI_BiaxialStage_SubID_1D_Y_Rep0: Rep4.csv | 1D-Y | pHRI_Manipulator_SubID_1D_Y_Rep0: Rep4.csv |
| 2D-DG | pHRI_BiaxialStage_SubID_2D_DG_Rep0: Rep4.csv pHRI_BiaxialStage_SubID_2D_DG_Session2_Rep0: Rep1.csv | 2D-XY | pHRI_Manipulator_SubID_2D_XY_Rep0: Rep4.csv |
| 2D-SQ | pHRI_BiaxialStage_SubID_2D_SQ_Rep0: Rep4.csv | 2D-YZ | pHRI_Manipulator_SubID_2D_YZ_Rep0: Rep4.csv |
| 2D-DM | pHRI_BiaxialStage_2D_SubID_DM_Rep0: Rep4.csv | 2D-XZ | pHRI_Manipulator_SubID_2D_XZ_Rep0: Rep4.csv |
| 2D-SQ-Diff-Size | pHRI_BiaxialStage_2D_SubID_SQ_diffSize_Rep0: Rep15.csv | 3D-XYZ | pHRI_Manipulator_SubID_3D_XYZ_Rep0: Rep4.csv |
| | | HRC in 3D-XYZ | pHRI_Manipulator_SubID_3D_XYZ_HRC_Rep0: Rep4.csv |

**Table 3.** Dataset summary.

| pHRI | 1D | 2D | 3D |
|---|---|---|---|
| **Biaxial Stage** | Total files: 120 | Total files: 186 | NA |
| Participant: S1-S17 Upper arm & Forearm FMG data | Col 1 # $F_x$/$F_y$ data Col 2:33 # FMG data | Col 1:2 # $F_x$, $F_y$ data Col 3:34 # FMG data | |
| **Manipulator** | Total files: 15 | Total files: 15 | Total files: 10 |
| Participant: S18 Forearm FMG data | Col 1:3 # $F_x$, $F_y$, $F_z$ data Col 4:19 # FMG data | Col 1:3 # $F_x$, $F_y$, $F_z$ data Col 4:19 # FMG data | Col 1:3 # $F_x$, $F_y$, $F_z$ data Col 4:19 # FMG data |

## 5. Discussion

This article describes the datasets that were collected during the studies conducted in [6–10]. A brief description of each study is provided below for readers' clarity and ease of understanding the relevance of the datasets.

In [6], interactive force estimation to manipulate the linear robot/ biaxial stage was derived from 32 FMG channels in a 2D planar workspace. Interactions occurred in five different dynamic motions (1D-X, 1D-Y, 2D-diagonal/DG, 2D-square/SQ and 2D-diamond/DM) to examine human intentions of manipulating the robot in an intended direction. The 2D motions required more gradually complex muscle activities and arm movements. Real-time evaluations conducted via intra-session (i.e., training and testing in the same session) supervised models (support vector regressor, SVR and kernel ridge regressor, KRR), and were found effective in real-time force estimations ($R^2$: 90–94% in 1D and 82–91% in 2D motions). In this study, a separate trained model was required to estimate forces for each participant interacting in each motion.

For industrial pHRI applications, general applicability of a trained model for all workers was investigated in [7,8]. A large volume of 32-channel FMG source data collected during real-time interactions between several participants and the linear robot in selected motions (1D-X, 2D-diagonal/DG, 2D-square/SQ and 2D-square of different sizes/SQ-diffSize) were used for training. Real-time evaluations of a generalized model to recognize applied forces from 32-channel, out-of-distribution (OOD) target data were conducted. In [7], the supervised generalized model based on support vector regression (SVR) was evaluated for recognizing interactive forces in a new intended motion, or for a new participant ($R^2$: 90–94% [1D-X], 80–85% [2D-DG]). While in [8], a generalized model based on convolutional neural network (CNN) was found effective in recognizing unseen, similarly intended motion or a participant interacting daily in the same intended motion for ($R^2$: 88% [2D-SQ], 89% [2D-SQ-diffSize]).

In [9], a 3D-HRC task of moving a wooden rod in collaboration with the 7-DoF Kuka LBR IIWA 14 robot was investigated via a 16-channel FMG forearm band. Additionally, interactions with the Kuka robot were investigated to estimate grasping forces in a dynamic motion in the 1D, 2D and 3D workspaces via an intra-session CNN model. To improve model performance and to generalize with adequate training data, a large volume of source data (long-term data) collected during interactions with the biaxial stage was used. A cross-domain generalization (CDG) method was implemented for transferring knowledge between this unrelated source (2D-pHRI platform) and the target data (3D-pHRI platform). A pretrained model with CNN architecture performed better in simple 1D grasping interactions ($R^2$: 79–87%), while its performance slightly improved during collaborative task of moving the rod in 3D ($R^2$: ≈60–63%).

In [10], a study was conducted to address the real-world problem of unlabeled or inadequate training data. Obtaining enough training data, having more participants, or labeling all data were not possible with the Kuka robot. Therefore, the study focused on synthetic FMG data generation by implementing domain randomization technique using a CNN-based generative adversarial network (GAN). Knowledge learnt from the latent feature distributions was transferred via semi-supervised learning during intra-session test data evaluation. For this investigation, pHRI with the Kuka robot in 1D (X, Y and Z directions) was investigated using 16-channel forearm FMG signals. The proposed model performed ($R^2$: 77–84%) like the supervised model ($R^2$: 78–88%) with fewer labeled training datasets (only 25% were labeled) and a large volume of unlabeled synthetic FMG data (2.5 times more than the real data).

These studies revealed that FMG-based model generalization, domain adaptation, and cross-domain generalization were possible where a pretrained model was evaluated to estimate interactive forces in dynamic motions [7–9]. In [5], we also found that recognizing hand grasping with FMG data was feasible via a transfer learning technique even with an unrelated dataset, i.e., the pretrained Alexnet model. Hence, in the future, these data can be used in pretraining a transfer learning model for research or industrial applications of

either FMG-based or other sensory-based pHRI activities. In [10], we generated synthetic FMG data using a few real FMG data from this dataset using domain randomizations; the aforementioned transformation techniques can be utilized in real-life FMG data generation for research work when collecting data is not possible. There was room for improvements for model performances during a collaborative task with the Kuka robot. Therefore, the use of this dataset by others can enhance pHRI quality in safe collaborations with either 16-channel or 32-channel FMG signals.

These data were collected over a few years with different setups, corresponding studies were conducted, and results were published. We did not include data in each article because they would be mostly a fraction of the whole dataset and would require us to describe the data repeatedly. Collecting these human–robot interaction data required an expensive setup, experienced research personnel, time and effort in recruiting participants and longer hours for gathering these data. Thus, we expect this release will have a great impact on the research field.

The dataset discussed in Section 4.1 was collected before the pandemic and several participants voluntarily participated. The pandemic started before we could collect the interaction data described in Section 4.2 with the serial manipulator. These data were collected when restricted use of research areas opened. Working with human participants was strictly monitored to avoid health hazards, and it became difficult to have volunteers at that time. We recruited only one participant to collect interactive data with the Kuka robot. As the project timeline had also finished by the time pandemic ended, we had no option to engage more human participants.

## 6. Conclusions

Implementing pHRI with FMG data by learning human intentions is a state-of-the-art research area for industrial application. With traditional machine learning and recent deep learning techniques, the FMG-based human interactions with robots show potential for industrial applications. Due to limited resources, collecting HRI data is expensive and time consuming. As it is hard to find any datasets or repositories of myographic signals or any other bio signals related to HRI applications, we expect to fill a void in the field with the published research works and the corresponding data. Therefore, the release of these FMG-based pHRI data with two different robots will be useful in future studies of human intents of movements during collaborative tasks, and will benefit the research community.

## 7. User Notes

Dataset is readily available on Zenodo and can be downloaded at: https://doi.org/10.5281/zenodo.6632020 (accessed on 28 June 2022).

**Author Contributions:** U.Z. investigated and developed methodologies, designed the protocol, collected the FMG-based pHRI datasets, prepared the dataset and wrote the manuscript. C.M. supervised and conceptualized the project, contributed to the design of the protocol and methods, and participated in manuscript revisions. All authors have read and agreed to the published version of the manuscript.

**Funding:** This research was supported by the Natural Sciences and Engineering Research Council of Canada (NSERC), the Canadian Institutes of Health Research (CIHR), and the Canada Research Chair (CRC) program.

**Institutional Review Board Statement:** The study was conducted in accordance with the Declaration of Helsinki and approved by the Office of the Research Ethics of Simon Fraser University, Burnaby, BC, Canada.

**Informed Consent Statement:** Informed consent was obtained from all subjects involved in the study.

**Data Availability Statement:** Data can be freely downloaded at: https://doi.org/10.5281/zenodo.6632020 (accessed on 7 November 2022) under Creative Commons Attribution 4.0 International License. The corresponding author can be contacted in case of need.

**Acknowledgments:** We would like to thank the participants for their voluntary contributions and members of Menrva Research Group for assisting in this project.

**Conflicts of Interest:** The authors declare no conflict of interest. The principal investigator, Carlo Menon, and members of his research team have a vested interest in commercializing the technology tested in this study if it is proven to be successful, and may benefit financially from its potential commercialization.

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
