# Peer review of "Dataset on Force Myography for Human–Robot Interactions"

_data_

Round 1
Reviewer 1 Report
The main problem with the paper is that the link to download the data that is provided in the paper does not work. The domain seems to be on the market to be bought.
However, there are also other problems, it is not clear what can be the interest of a broader public for such data, that seems to me very specific to the setting in which they were collected. Moreover, the data seems to have been used for a number of publications, and I think that it would have been more useful to add to these publications the link to the data instead of having a standing-alone publication.
Author Response
Reviewer 1 (R1)
The article is well written with almost just some typing mistakes (marked in the pdf uploaded). In the following there are some points that should be changed in order to get a paper with better quality.
R1.1
The main problem with the paper is that the link to download the data that is provided in the paper does not work. The domain seems to be on the market to be bought.
Response:
We thank the reviewer for this valuable remarks.
Action:
Full link is provided in the manuscript:
Dataset: https://doi.org/10.5281/zenodo.6632020
R1.2
However, there are also other problems, it is not clear what can be the interest of a broader public for such data, that seems to me very specific to the setting in which they were collected. Moreover, the data seems to have been used for a number of publications, and I think that it would have been more useful to add to these publications the link to the data instead of having a standing-alone publication.
Response:
We thank the reviewer for the comments. We have revised Summary section and have included Discussion section followed by a Conclusion section.
Our justifications of publishing the data are discussed and included in these sections.
Action:
At Section: Summary:
Including human biosignals to control or interact with machines is an ongoing re-search area. Many studies used biosignals such as electroencephalogram (EEG), electrooculogram (EOG) and electromyogram (EMG) for neuronal association with machines. These biosignals are mostly acquired from a specialized tissue, organ, or the nervous system. Research have shown that including human feedback via one or more of these biosignals are feasible [20-26]. However, implementing human machine interactions via these signals are sometimes impractical because of poor signal quality, bulky signal processing equipment, and restrictive human movements. Alternatively, many HRI studies [27-34] utilized the traditional, non-invasive, wearable surface electromyography (sEMG) technique which measures electrical activities of underlying muscle bundles. However, none of these myography-based HRI datasets are publicly available.
Very recently, few FMG-based HRI studies were conducted by the authors of this article where human applied interaction force was predicted to perform collaborative task. There is no other similar research work in predicting human interactive forces for HRC task like these studies because of the uncertain and dynamic HRI environment, availability of research funding and trained personnel. Hence, this paper releases a dataset under the CC BY NC ND 4.0 license collected during the studies conducted in [6-10] while dynamic interactions happened between several human participants with a biaxial stage and a Kuka robot. These experimental data were collected in certain setups. A participant interacted with the robot applying hand forces in several dynamic arm movements that represented common activities during human robot interactions. So far, in our knowledge, there is not a publicly available FMG dataset on pHRI application. Hence, this dataset will provide a starting point for future research works and avoid the need of collecting data. These data will also help researchers to understand the nature of the sensory signals that captured muscle activities during certain pHRI interactions. Our goal is to provide insight about the FMG signals, its applicability as a safety band in human-robot interactions and inspire other researchers to work on this dataset.
At Section: Discussion:
These data were collected over few years with different setups, corresponding studies were conducted, and results were published. We did not include data in each article because it would be mostly a fraction of the whole dataset and would require us to describe the data repeatedly.
In our studies, we showed that FMG-based model generalization, domain adaptation, and cross-domain generalization were possible where a pretrained model was evaluated to estimate interactive forces in dynamic motions [7-9]. In [5], we found that recognizing hand grasping with FMG data was doable via transfer learning technique even with unrelated dataset, i.e., the pretrained Alexnet model. Hence, in future, these data can be used in pretraining a transfer learning model for research or industrial applications of either FMG-based or other sensory-based pHRI activities. In [10], we generated synthetic FMG data from few real FMG data from this dataset using domain randomizations; the mentioned transformation techniques can be utilized in real-like FMG data generation for research work when collecting data is not possible.

Reviewer 2 Report
This study proposed a dataset which contains FMG data collected from two separate pHRI setups: S1-S17 participants interacted with a biaxial stage, and S18 participant with a serial manipulator. The dataset will contribute to the research in biosignal-based HRI control design. There are some minor comments:
1. The tittle is too broad, and it should be appropriately modified according to the research results of this manuscript, such as including the key words"FMG-based pHRI dataset".
2. Is gender as a factor considered when constructed the data set? Does gender affect the pHRI in the Biaxial Stage and a Manipulator?
3. In Line 136 , "Participant S6 grabbed the open-end..." but in Figure 5(b) and Line 189, the Participant is S18, please give more detailed instruction.
4. Please explain in the manuscript why the biaxial stage needs 17 participants, while the serial manipulator only has 1 participant as sample. Is the data thus obtained comparable? Moreover, the data obtained in the experiment with the serial manipulator would have great uncertainty. Can it be representative?
Author Response
Reviewer 2 (R2)
This study proposed a dataset which contains FMG data collected from two separate pHRI setups: S1-S17 participants interacted with a biaxial stage, and S18 participant with a serial manipulator. The dataset will contribute to the research in biosignal-based HRI control design. There are some minor comments:
R2.1
The tittle is too broad, and it should be appropriately modified according to the research results of this manuscript, such as including the key words "FMG-based pHRI dataset".
Response:
We thank the reviewer for this comment.
We elaborated the term ‘FMG’ and ‘HRI in the title for broader reference so that the readers can easily and quickly search and obtain the article and the dataset from online sources.
R2.2
Is gender as a factor considered when constructed the data set? Does gender affect the pHRI in the Biaxial Stage and a Manipulator?
Response:
We thank the reviewer for the valuable comment.
During interactions with the biaxial stage, three (3) female participants were engaged. We did not see any variations among the various ages of male and female performances. This was discussed in [6]. We have included a plot of FMG signals capturing muscle readings during interactions with the biaxial stage for a male and a female participant, as shown in Figure 6.
|
(a) (b) Figure 6. 32-channel FMG signals capturing muscle readings during interactions with biaxial stage in DG motion: a) A male participant, and b) A female participant. |
R2.3
In Line 136 , "Participant S6 grabbed the open-end..." but in Figure 5(b) and Line 189, the Participant is S18, please give more detailed instruction.
Response:
We are grateful to the reviewer for reviewing meticulously.
It was an honest typo, and we are grateful that we could correct the mistake.
Action:
S6 was replaced with S18 in the manuscript.
R2.4
Please explain in the manuscript why the biaxial stage needs 17 participants, while the serial manipulator only has 1 participant as sample. Is the data thus obtained comparable? Moreover, the data obtained in the experiment with the serial manipulator would have great uncertainty. Can it be representative?
Response:
We thank the reviewer for this valuable comment.
We had the opportunity in the first phase of the FMG-based HRI project to have more participants to interact with the biaxial stage. However, pandemic started during the second phase of the project. We were prohibited conducting any data collection with human participants to interact with the serial manipulator for a long period. When human involved research finally resumed in the university premises, we could only have one participant before the project terminated.
Since we had only one participant for collaborative interactions with the Kuka robot, we collected all possible interaction data in 1D, 2D, and 3D. We showed that interaction data from other participants could become useful for domain generalization [9]. With transfer learning, we predicted interactive applied forces with the Kuka robot for the only one participant using 5 other participants’ interactive data with the biaxial stage. And we conducted a study when data could be scarce in [10], as was the case for us and showed how to face such challenges with limited no. of participant or limited data.
Action:
At Section: Discussion:
The dataset discussed in Section 4.1 were collected before the pandemic and several participants voluntarily participated. The pandemic started before we could collect inter-action data described in Section 4.2 with the serial manipulator. These data were collected when restricted use of research area opened, working with human participant was strictly monitored to avoid health hazards, and it became difficult to have volunteers at that time. We could recruit only one participant to collect interactive data with the Kuka robot. As the project timeline also finished by the time pandemic ended, we had no option to hire more human participants.
In our studies, we showed that FMG-based model generalization, domain adaptation, and cross-domain generalization were possible where a pretrained model was evaluated to estimate interactive forces in dynamic motions [7-9]. In [5], we found that recognizing hand grasping with FMG data was doable via transfer learning technique even with unrelated dataset, i.e., the pretrained Alexnet model. Hence, in future, these data can be used in pretraining a transfer learning model for research or industrial applications of either FMG-based or other sensory-based pHRI activities. In [10], we generated synthetic FMG data from few real FMG data from this dataset using domain randomizations; the mentioned transformation techniques can be utilized in real-like FMG data generation for research work when collecting data is not possible.

Reviewer 3 Report
In this paper, the authors present a database of FMG signals of 16-channel (forearm) and 32-channel (forearm and upper arm) bands while participants interacted with a biaxial stage (linear robot) and a serial manipulator (Kuka robot. The authors argue that the dataset can be useful in future studies on FMG biosignal-based pHRI control design.
While the paper is understandable, it will be helpful to proofread it carefully as there are small issues here and there and there are statements that are not that clear (e.g., lines 27-29, "In literature", etc.).
In addition, it will be helpful for the authors to clarify the following:
"Understanding human intended intentions of interactions can be vital in developing safe collaborations between a human worker and a manipulator" How is the dataset help with this? The introduction does not make it clear how intention is captured by the data and how this data can be used to interpret users' intentions.
Figure examples of the signals from the FMG hands will be helpful to understand the differences between the data signals.
All figures have low resolution and are hence not so clear. It will be better to have high-resolution figures.
Datasets 1 and 2. The authors provide very short descriptions of data collection and results of the experiment analysis and point to other papers. More information should be provided in the current paper, in a summarized form; otherwise, this paper is not self-contained. It is fine if one can get a good understanding but this does not seem to be the case.
About the conflict of interest statement, the two sentences seem to contradict each other. It will be good for the authors to clarify them.
In short, as this paper is about a dataset, it is important for the authors to add more information to the paper to allow readers to have a good understanding of the importance of the dataset and what the data is analyzed (examples will be helpful).
Author Response
Reviewer 3 (R3)
In this paper, the authors present a database of FMG signals of 16-channel (forearm) and 32-channel (forearm and upper arm) bands while participants interacted with a biaxial stage (linear robot) and a serial manipulator (Kuka robot. The authors argue that the dataset can be useful in future studies on FMG biosignal-based pHRI control design.
R3.1
While the paper is understandable, it will be helpful to proofread it carefully as there are small issues here and there and there are statements that are not that clear (e.g., lines 27-29, "In literature", etc.).
Response:
We thank the reviewer for this comment. We have emphasized on better user readability and proofread the paper carefully.
R3.2
In addition, it will be helpful for the authors to clarify the following:
"Understanding human intended intentions of interactions can be vital in developing safe collaborations between a human worker and a manipulator" How is the dataset help with this? The introduction does not make it clear how intention is captured by the data and how this data can be used to interpret users' intentions.
Response:
We thank the reviewer for the valuable comment.
We have revised the Introduction section and included new sections: Discussions, Limitations, and Conclusions.
Action:
Included in the Summary section:
Very recently, few FMG-based HRI studies were conducted by the authors of this article where human applied interaction force in dynamic motion was predicted to perform collaborative task. There is no other similar research work in predicting human interactive forces for HRC task like these studies because of the uncertain and dynamic HRI environment, availability of research funding and trained personnel. Understanding human intended intentions of interactions can be vital in developing safe collaborations between a human worker and a manipulator. Hence, this paper releases a dataset under the CC BY NC ND 4.0 license collected during the studies conducted in [6-10] while dynamic interactions happened between several human participants with a biaxial stage and a Kuka robot. These experimental data were collected in certain setups. A participant interacted with the robot applying hand forces in several dynamic arm movements that represented common activities during human robot interactions. The FMG bands wrapped around the forearm, or the upper arm, captured the muscle readings of a participant during an activity with the robot and were mapped to applied force in a certain motion via a trained model. So far, in our knowledge, there is not a publicly available FMG dataset on pHRI application. Hence, this dataset will provide a starting point for future research works and avoid the need of collecting data. These data will also help researchers to understand the nature of the sensory signals that captured muscle activities during certain pHRI interactions. Our goal is to provide insight about the FMG signals, its applicability as a safety band in human-robot interactions and inspire other researchers to work on this dataset. As the studies on FMG-based pHRI have revealed its viability [6-10], we hope that this release will aid to fill the gap in available datasets and to facilitate future research in biosignal-based HRI control design.
R3.3
All figures have low resolution and are hence not so clear. It will be better to have high-resolution figures.
Response:
We thank the reviewer.
These pictures were taken during the project work and not possible to produce these again.
R3.4
Datasets 1 and 2. The authors provide very short descriptions of data collection and results of the experiment analysis and point to other papers. More information should be provided in the current paper, in a summarized form; otherwise, this paper is not self-contained. It is fine if one can get a good understanding but this does not seem to be the case.
Response:
We thank the reviewer for this valuable suggestion.
We have added Discussion section and have summarized the published articles related to the dataset.
Action:
At Section: Discussion:
This article describes the datasets that were collected for conducting the studies in [6-10]. A brief description of each study is provided below for readers’ clarity and ease of understanding the relevance.
In [6], interactive force estimation to manipulate the linear robot/ biaxial stage was derived from 32 FMG channels in 2D planar workspace. Interactions occurred in five different dynamic motions (1D-X, 1D-Y, 2D-diagonal/DG, 2D-square/SQ and 2D-diamond/DM) to examine human intentions of manipulating the robot in an intended di-rection. The 2D motions required gradually complex muscle activities and arm movements. Real-time evaluations conducted via intra-session (i.e., training and testing in the same session) supervised models (support vector regressor, SVR and kernel ridge regressor, KRR) were found effective in real-time force estimations (R2: 90%- 94% in 1D and 82% - 91% in 2D motions). In this study, a separate trained model was required for each participant in each motion.
For industrial pHRI application, general applicability of a trained model for all workers was investigated in [7][8]. A large volume of 32-channel FMG source data collect-ed during real-time interactions between several participants and the linear robot in selected motions (1D-X, 2D-diagonal/DG, 2D-sqaure/SQ and 2D-sqaure of different sizes/SQ-diffSize) were used for training. Real-time evaluations of a generalized model to recognize applied forces from 32-channel, out-of-distribution (OOD) target data were conducted. In [7], the supervised generalized model based on support vector regression (SVR) was evaluated for recognizing interactive forces in a new intended motion or for a new participant (R2: 90%- 94% [1D-X], 80%-85% [2D-DG]. While in [8], a generalized model based on convolutional neural network (CNN) was found effective in recognizing unseen, similar in-tended motion or a repetitive participant interacting daily in same intended motion for (R2: 88% [2D-SQ], 89% [2D-SQ-diffSize]).
In [9], a 3D-HRC task of moving a wooden rod in collaboration with the 7-DoF Kuka LBR IIWA 14 robot was investigated via 16-channel FMG forearm band. Also, interactions with the Kuka robot were investigated to estimate grasping forces in dynamic motion in the 1D, 2D and 3D workspaces via an intra-session CNN model. To improve model performance and to generalize with adequate training data, large volume of source data (long-term data) from the interactions with the biaxial stage was used. A cross-domain generalization (CDG) method was implemented for transferring knowledge between this unrelated source (2D-pHRI platform) and the target data (3D-pHRI platform). A pre-trained model with CNN architecture performed better in simple 1D grasping interactions (R2: 79-87%) while its performance slightly improved during collaborative task of moving the rod in 3D (R2: ≈60-63%).
In [10], a study was conducted to address the real-world problem of unlabeled or in-adequate training data. Obtaining enough training data, having more participants, or labeling all data were not possible with the Kuka robot. Therefore, the study focused on syn-thetic FMG data generation by implementing domain randomization technique using a CNN-based generative adversarial network (GAN). Knowledge from the latent feature distributions was transferred via semi-supervised learning during intra-session test data evaluation. For this investigation, pHRI with the Kuka robot in 1D (X, Y and Z directions) was investigated using 16-channel forearm FMG signals. The proposed model performed (R2: 77%-84%) like the supervised model (R2: 78%-88%) even with fewer labeled training datasets (such as labeled vs unlabeled = 1:4) and a large volume of unlabeled, generated synthetic FMG data (real vs syn. = 1: 2.5).
The studies revealed that FMG-based model generalization, domain adaptation, and cross-domain generalization were possible where a pretrained model was evaluated to estimate interactive forces in dynamic motions [7-9]. In [5], we found that recognizing hand grasping with FMG data was doable via transfer learning technique even with unrelated dataset, i.e., the pretrained Alexnet model. Hence, in future, these data can be used in pre-training a transfer learning model for research or industrial applications of either FMG-based or other sensory-based pHRI activities. In [10], we generated synthetic FMG data from few real FMG data from this dataset using domain randomizations; the mentioned transformation techniques can be utilized in real-like FMG data generation for research work when collecting data is not possible. There is room for improvements for model performances during collaborative task with the Kuka robot. Therefore, the use of this dataset by others can enhance pHRI quality in safe collaborations with either 16-channel or 32-channel FMG signals.
R3.5
About the conflict of interest statement, the two sentences seem to contradict each other. It will be good for the authors to clarify them.
Response:
We thank the reviewer for the comment.
We have no conflict of interest at this point. The principal investigator and his research team have interest in commercializing the technology in future if possible.
R3.6
In short, as this paper is about a dataset, it is important for the authors to add more information to the paper to allow readers to have a good understanding of the importance of the dataset and what the data is analyzed (examples will be helpful).
Response:
We thank the reviewer for the comment.
We have revised the Summary section to justify the need of releasing the dataset, added Discussion section have summarized the published articles related to the dataset for describing the published articles and how FMG data was interpreted to recognize interactive force in dynamic motions, and concluded in Conclusion section.
We have addressed the concerns of the reviewer throughout the revised manuscript for readers’ clarity in ease of understanding the association of the articles and the dataset to its relevance.
Action:
Included in the Summary section:
Including human biosignals to control or interact with machines is an ongoing research area. Many studies used biosignals such as electroencephalogram (EEG), electrooculogram (EOG) and electromyogram (EMG) for neuronal association with machines. These biosignals are mostly acquired from a specialized tissue, organ, or the nervous system. Research have shown that including human feedback via one or more of these biosignals were feasible [20-26]. However, implementing human machine interactions via these signals are sometimes impractical because of poor signal quality, bulky signal processing equipment, and restrictive human movements. Alternatively, many HRI studies [27-34] utilized the traditional, non-invasive, wearable surface electromyography (sEMG) technique which measures electrical activities of underlying muscle bundles. However, none of the myography-based HRI datasets are publicly available. Interestingly, the FMG technique was found comparable with the conventional sEMG technique in hand gesture recognition, rehabilitation or prosthetic control applications or human machine interaction studies. Research has shown that the FMG was advantageous compared to sEMG technique with lower cost, smaller signal processing units using Bluetooth technology, ease of wearing the band and, hence, a better choice for HMI [35-37]. Recently, FMG-based human robot collaboration (HRC) tasks were conducted where an industrial YUMI robot avoided collision with a human participant by recognizing intentional or random hand movements [38]. In separate HRC studies, grasping forces via FMG data was used to recognize the intended tool (object) grasped by the human worker during a shared task [39-40].
Very recently, a few FMG-based HRI studies were conducted by the authors of this article where human applied interaction force in dynamic motion was predicted to perform a collaborative task. There is no other similar research work in predicting human interactive forces for HRC tasks like these studies because of the uncertain and dynamic HRI environment, availability of research funding and trained personnel. Understanding human intended intentions of interactions can be vital in developing safe collaborations between a human worker and a manipulator. Hence, this paper releases a dataset under the CC BY NC ND 4.0 license collected during the studies conducted in [6-10] while dynamic interactions happened between several human participants with a biaxial stage and a Kuka robot. These experimental data were collected using certain setups. A participant interacted with the robot applying hand forces in several dynamic arm movements that represented common activities during human robot interactions. The FMG bands wrapped around the forearm, or the upper arm, captured the muscle readings of a participant during an activity with the robot and were mapped to an applied force in a certain motion via a trained model. So far, to our knowledge, there is not a publicly available FMG dataset on pHRI application. Hence, this dataset will provide a starting point for future research works and avoid the need for collecting data. These data will also help researchers to understand the nature of the sensory signals that captured muscle activities during certain pHRI interactions. Our goal is to provide insight about the FMG signals, its applicability as a safety band in human-robot interactions and inspire other researchers to work on this dataset. As the studies on FMG-based pHRI have revealed its viability [6-10], we hope that this release will aid to fill the gap in available datasets and to facilitate future research in biosignal-based HRI control design.
Added Figure 6 at Section 4.1. Dataset 1: pHRI_Biaxial Stage:
|
(a) (b) Figure 6. 32-channel FMG signals capturing muscle readings during interactions with biaxial stage in DG motion: a) A male participant, and b) A female participant. |
Figure 6 shows the FMG signals capturing interactions with the biaxial stage in diagonal motion for a male and a female participant. Plot of these captured signals show that the participants were interacting with the stage with their own pace of applied force and arm motions.
Added Figure 7 at Section 4.2. Dataset 2: pHRI_Manipulator:
Figure 7 shows a few plots of the FMG signals during interactions with the Kuka robot in a certain direction in 1D-X, Y, and 3D-XYZ.
|
(a) (b)
(c) Figure 7. pHRI between participant S18 and the Kuka robot: a) in 1D-X, and (b) in 1D-Y, and c) in 3D for HRC task. Reproduced from [9] © [2022] IEEE with permission.
|
Added a new Section: Discussion:
This article describes the datasets that were collected for conducting the studies in [6-10]. A brief description of each study is provided below for readers’ clarity and ease of understanding the relevance.
In [6], interactive force estimation to manipulate the linear robot/ biaxial stage was derived from 32 FMG channels in 2D planar workspace. Interactions occurred in five different dynamic motions (1D-X, 1D-Y, 2D-diagonal/DG, 2D-square/SQ and 2D-diamond/DM) to examine human intentions of manipulating the robot in an intended di-rection. The 2D motions required gradually complex muscle activities and arm movements. Real-time evaluations conducted via intra-session (i.e., training and testing in the same session) supervised models (support vector regressor, SVR and kernel ridge regressor, KRR) were found effective in real-time force estimations (R2: 90%- 94% in 1D and 82% - 91% in 2D motions). In this study, a separate trained model was required for each participant in each motion.
For industrial pHRI application, general applicability of a trained model for all workers was investigated in [7][8]. A large volume of 32-channel FMG source data collect-ed during real-time interactions between several participants and the linear robot in selected motions (1D-X, 2D-diagonal/DG, 2D-sqaure/SQ and 2D-sqaure of different sizes/SQ-diffSize) were used for training. Real-time evaluations of a generalized model to recognize applied forces from 32-channel, out-of-distribution (OOD) target data were conducted. In [7], the supervised generalized model based on support vector regression (SVR) was evaluated for recognizing interactive forces in a new intended motion or for a new participant (R2: 90%- 94% [1D-X], 80%-85% [2D-DG]. While in [8], a generalized model based on convolutional neural network (CNN) was found effective in recognizing unseen, similar in-tended motion or a repetitive participant interacting daily in same intended motion for (R2: 88% [2D-SQ], 89% [2D-SQ-diffSize]).
In [9], a 3D-HRC task of moving a wooden rod in collaboration with the 7-DoF Kuka LBR IIWA 14 robot was investigated via 16-channel FMG forearm band. Also, interactions with the Kuka robot were investigated to estimate grasping forces in dynamic motion in the 1D, 2D and 3D workspaces via an intra-session CNN model. To improve model performance and to generalize with adequate training data, large volume of source data (long-term data) from the interactions with the biaxial stage was used. A cross-domain generalization (CDG) method was implemented for transferring knowledge between this unrelated source (2D-pHRI platform) and the target data (3D-pHRI platform). A pre-trained model with CNN architecture performed better in simple 1D grasping interactions (R2: 79-87%) while its performance slightly improved during collaborative task of moving the rod in 3D (R2: ≈60-63%).
In [10], a study was conducted to address the real-world problem of unlabeled or in-adequate training data. Obtaining enough training data, having more participants, or labeling all data were not possible with the Kuka robot. Therefore, the study focused on syn-thetic FMG data generation by implementing domain randomization technique using a CNN-based generative adversarial network (GAN). Knowledge from the latent feature distributions was transferred via semi-supervised learning during intra-session test data evaluation. For this investigation, pHRI with the Kuka robot in 1D (X, Y and Z directions) was investigated using 16-channel forearm FMG signals. The proposed model performed (R2: 77%-84%) like the supervised model (R2: 78%-88%) even with fewer labeled training datasets (such as labeled vs unlabeled = 1:4) and a large volume of unlabeled, generated synthetic FMG data (real vs syn. = 1: 2.5).
The studies revealed that FMG-based model generalization, domain adaptation, and cross-domain generalization were possible where a pretrained model was evaluated to estimate interactive forces in dynamic motions [7-9]. In [5], we found that recognizing hand grasping with FMG data was doable via transfer learning technique even with unrelated dataset, i.e., the pretrained Alexnet model. Hence, in future, these data can be used in pre-training a transfer learning model for research or industrial applications of either FMG-based or other sensory-based pHRI activities. In [10], we generated synthetic FMG data from few real FMG data from this dataset using domain randomizations; the mentioned transformation techniques can be utilized in real-like FMG data generation for research work when collecting data is not possible. There is room for improvements for model performances during collaborative task with the Kuka robot. Therefore, the use of this dataset by others can enhance pHRI quality in safe collaborations with either 16-channel or 32-channel FMG signals.
Added Conclusion section:
Implementing pHRI with FMG data by learning human intentions is a state-of-the-art research area for industrial application. With traditional machine learning and recent deep learning techniques, the FMG-based human interactions with robots shows potential for the use in industries using pHRI. Due to limited resources, collecting HRI data is expensive and time consuming. As it is hard to find any datasets or repositories of myo-graphic signals or any other bio signals related to HRI applications, we expect to fulfill a void in the field by the published research works and the corresponding data. Therefore, the release of these FMG-based pHRI data with two different robots will be useful in future studies of human intents of movements during collaborative tasks and benefit the re-search community.

Reviewer 4 Report
The dataset in this paper is very conducive to the research of physical human robot interactions. Thanks to the authors for collecting and making the dataset available.
The manuscript is generated in the revision mode, which affects reading, and it is hoped that it will be checked carefully before submission in the future.
Author Response
We thank the reviewer for the comment and suggestion.
The final manuscript is revised and checked for grammetical errors and typos.
Round 2
Reviewer 3 Report
Overall, this paper is much improved. The authors did a good job in revising their submission. This paper fits the scope of the journal and is aligned with the presentation of this type of work. While it the interest in this work may be limited, it has merits in its publication.
Author Response
We thank the reviewer for the nice comment.
We have proofread the manuscript for grammetical and/or spell check.